# Exploring the Characteristic Aroma Components of Traditional Fermented Koumiss of Kazakh Ethnicity in Different Regions of Xinjiang by Combining Modern Instrumental Detection Technology with Multivariate Statistical Analysis Methods for Odor Activity Value and Sensory Analysis

**DOI:** 10.3390/foods12112223

**Published:** 2023-05-31

**Authors:** Yongzhen Gou, Xinmiao Ma, Xiyue Niu, Xiaopu Ren, Geminguli Muhatai, Qian Xu

**Affiliations:** College of Food Science and Engineering, Tarim University, Alar 843301, China; 19190316975@163.com (Y.G.);

**Keywords:** koumiss, GC-IMS, GC-MS, PLS-DA, OVA, volatile components, QDA, origin discrimination, aroma-presenting substances

## Abstract

To investigate the characteristic aromatic compounds, present in the traditional fermented koumiss of the Kazakh ethnic group in different regions of Xinjiang, GC-IMS, and GC-MS were used to analyze the volatile compounds in koumiss from four regions. A total of 87 volatile substances were detected, and esters, acids, and alcohols were found to be the main aroma compounds in koumiss. While the types of aroma compounds in koumiss were similar across different regions, the differences in their concentrations were significant and displayed clear regional characteristics. The fingerprint spectrum of GC-IMS, combined with PLS-DA analysis, indicates that eight distinctive volatile compounds, including ethyl butyrate, can be utilized to distinguish between different origins. Additionally, we analyzed the OVA value and sensory quantification of koumiss in different regions. We found that aroma components such as ethyl caprylate and ethyl caprate, which exhibit buttery and milky characteristics, were prominent in the YL and TC regions. In contrast, aroma components such as phenylethanol, which feature a floral fragrance, were more prominent in the ALTe region. The aroma profiles of koumiss from the four regions were defined. These studies provide theoretical guidance for the industrial production of Kazakh koumiss products.

## 1. Introduction

The vast grasslands in northern Xinjiang, where the developed animal husbandry industry and ancient nomadic Kazakh people reside, have nurtured a long history of making and consuming koumiss [1]. Some traditional fermented dairy products are still an indispensable part of people’s lives today. Fresh mare’s milk is fermented under specific temperatures and time through the joint action of lactic acid bacteria, yeast, and other microorganisms to form koumiss [2]. Koumiss has high nutritional value [3], can effectively improve intestinal flora [4], and has certain auxiliary therapeutic effects on cardiovascular disease, gastrointestinal infections, diabetes, and tuberculosis [5]. Xinjiang Kazakh koumiss differs from the Western industrialized koumiss [6] and relies on traditional natural fermentation, which has a unique fermentation process. The flavor and sensory attributes of Kazakh koumiss are influenced by factors, such as fermentation agents, regional conditions, and fermentation process [7]. Koumiss in different regions has its own characteristics, forming unique sensory properties and flavors.

In recent years, with the burgeoning development of the equine industry in Xinjiang, koumiss, as a high-value-added food, has gradually transformed from a mere pastoral self-sustenance beverage to a constantly selling commodity [8]. With the improvement of people’s economic level, the demand for food with medical and health benefits is increasing [9], and the market prospects for koumiss are also becoming broader. However, the uncertainty of the flavor of koumiss in different regions has become a bottleneck that restricts the development of the koumiss industry. Due to the differences in the composition and content of substances in koumiss in different regions, the differences in flavor and quality of koumiss are determined [10]. Therefore, in-depth research on these substances can guide the targeted processing and flavor quality control of koumiss products, thereby producing corresponding economic benefits. This will also help promote local horse industry economic growth, assist in rural revitalization strategies, and increase the income of herdsmen, which has important economic and social significance.

Gas chromatography–ion mobility spectrometry (GC-IMS) is an emerging detection technology in food aroma detection, which has the advantages of high sensitivity and strong convenience and is widely used in food quality control and traceability [11]. Zhang et al. [12] analyzed the volatile compounds of different years of brewed liquor using GC-IMS and chemometrics. By comparing the differences in the content of volatile compounds, fingerprint spectra were quickly generated, and rapid differentiation of different years of brewed liquor was achieved through principal component analysis (PCA). Liu et al. [13] analyzed the fermented milk of *Lactobacillus delbrueckii subsp. bulgaricus* of cheese type and fermentation type by GC-IMS and flavoromics, finally screening out six volatile compounds, such as 2,3-butanedione, as key aroma active substances. Literature studies have shown that the application of GC-IMS in fermented dairy products is in its preliminary stage, and there are almost no literature reports on the application of GC-IMS in koumiss. Therefore, this study innovatively uses GC-IMS to analyze the changes in volatile substances in koumiss in different regions.

In recent years, research on the aroma of koumiss has been gradually developing. Meng and colleagues [14], through their study of the microbial composition and aroma changes during the fermentation process of koumiss in Xilingol League, Inner Mongolia, found that the core microbial groups responsible for producing volatile flavor compounds in koumiss are lactobacillus, Brettanomyces, and Torulaspora. Additionally, the formation of koumiss aroma depends on the type and amount of acid, ester, and alcohol present. Xia and colleagues [15] used high-throughput sequencing and liquid-liquid extraction-gas chromatography-mass spectrometry to analyze the time-dependent changes in microbial groups and volatile compounds in koumiss from Xilingol League, Inner Mongolia. They identified lactobacillus and Dekkera as the core microorganisms in koumiss. During the fermentation process of koumiss, 52 volatile compounds were identified, including 15 different metabolic products of volatile compounds. These showed a significant positive correlation with the synthetic metabolism of lactobacillus and Dekkera.

To our knowledge, the aroma compounds of traditional fermented Kazakh-style koumiss from different regions in Xinjiang have not been systematically studied. This study focuses on the traditional fermented koumiss of the Kazakh ethnic group in four different regions of Xinjiang. By utilizing the convenient and rapid GC-IMS technique and employing multivariate statistical analysis, we have established fingerprint profiles to differentiate the volatile compounds of koumiss from different regions. In tandem with gas chromatograph-mass spectrometer (GC-MS) analysis, utilizing volatile compounds as the foundation, we have integrated odor activity value (OAV) analysis and quantitative descriptive analysis (QDA) based on a sensory evaluation to characterize the sensory aromatic attributes of koumiss from various regions. This study provides a theoretical basis for understanding the volatile substance and sensory characteristics of traditional fermented Kazakh-style koumiss in Xinjiang and improving its quality.

## 2. Materials and Methods

### 2.1. Sample Collection and Material Reagents

The samples were collected in early July 2022 from the households of Kazakh farmers and herdsmen in four regions (Table 1) surrounding the Altai and Tianshan Mountains in northern Xinjiang, China. To ensure the representative significance of samples from different regions, the linear distance between sampling locations is guaranteed to exceed 350 km. The collected samples were stored in sealed containers and transferred to the laboratory, where they were kept at −80 °C in liquid nitrogen for subsequent experiments. The fermentation methods of the koumiss samples from the four regions varied, as shown in Figure 1. Chemicals used in the study included 3-octanol (Aladdin Biochemical Technology Co., Ltd., Shanghai, China), normal ketones C4~C9 (China National Pharmaceutical Chemical Reagent Co, Ltd., Beijing, China), and normal alkanes (C7~C30) Sigma-Aldrich (St. Louis, MO, USA). All chemicals used were of chromatographic grade purity.

### 2.2. Analysis by Gas Chromatography-Ion Mobility Spectrometry

2.0 mL of koumiss sample was introduced into a 20 mL headspace vial (Agilent Technologies, Santa Clara, CA, USA). A 10 μL aliquot of 3-octanol solution (300 mg/L) was added to the sample, followed by incubation at 50 °C with stirring at 500 rpm for 15 min. After the incubation, 500 μL of headspace gas was automatically extracted without splitting into the GC-IMS instrument (FlavorSpec, GAS, Dortmund, Germany). The sample was separated using a WAT-Wax capillary column (15 m × 0.53 mm) with a temperature of 60 °C. Nitrogen gas (99.99% purity) was used as the carrier gas. The flow rate was controlled as follows: initially kept at 2 mL/min for 2 min, increased to 10 mL/min over 8 min, increased to 100 mL/min over 5 min, and finally increased to 150 mL/min over 5 min and kept constant for 10 min. The drift tube was 98.0 mm long, with a working temperature of 45 °C and a constant voltage of 5 kV. The entire analysis took 45 min, and all analyses were performed in triplicate.

The GC-IMS data obtained were analyzed using the laboratory analytical viewer (LAV). The Reporter plugin was used to directly compare the two-dimensional top view and difference spectra, while the gallery plot was used for fingerprint spectrum comparison. The retention index (RI) of the volatile compounds was calculated using the internal standard of normal ketones C4–C9. The identified volatile compounds were compared with the IMS database (G.A.S, Dortmund, Germany) using the retention index (RI), normalized drift time (RIP), and other parameters.

### 2.3. Gas Chromatography-Mass Spectrometry Analysis

8 g of koumiss sample was placed in a 25 mL headspace vial (Agilent Technologies) with a silicon septum, and 200 μL of 300 mg/L 3-octanol solution was added. The sample was equilibrated at 65 °C for 30 min. A solid-phase microextraction fiber coated with 50/30 μm divinylbenzene/carboxen/polydimethylsiloxane (DVB/CAR/PDMS) (Shanghai ANPEL Laboratory Technologies Co., Ltd., Shanghai, China) was inserted into the headspace of the glass vial for 30 min at 65 °C. After extraction, the fiber was inserted into the inlet of the GC-MS and thermally desorbed for 3 min at 25 °C. The GC-MS analysis was performed using an Agilent 7890B–7000D gas chromatography-mass spectrometry instrument, and volatile compounds was separated on a DB-WAX column (60 mm × 0.25 mm, 0.25 μm, Agilent). The temperature program started with a 5 min hold at 40 °C, followed by an increase to 200 °C at a rate of 3 °C/min, and a final increase to 250 °C at a rate of 6 °C/min. The total analysis time for each sample was 122 min. The retention indices (RI) were determined using normal alkane hydrocarbons (C7~C30), and volatile compounds were identified by mass spectrometry and RI values.

### 2.4. Quantitative Descriptive Analysis (QDA)

Following the method outlined in ISO 8586:2012 (ISO, 2012) [16], 20 students and faculty members from the School of Food Science and Engineering at Tarim University in China were selected as sensory panelists. These individuals received relevant training and had previous experience in sensory analysis. Over one month, with an average of one hour per day, the 20 panelists underwent training to familiarize themselves with and identify the common odors in fermented dairy products. Based on their sensory perceptual abilities, 10 panelists (4 male and 6 female; average age of 24 years) were selected to undergo additional sensory training, where they were required to detect standard reference materials and grade the intensity of different aroma attributes. Further, the training and selection of the panelists, as well as the subsequent sensory descriptive analysis, were conducted in the sensory laboratory at the School of Food Science and Engineering at Tarim University, with room temperature maintained at 20 °C.

According to ISO 5492:2008 (ISO, 2008) [17], six sensory descriptive terms including fruitiness, creaminess, floral, off-flavor, dairy flavor, and sourness were selected to measure the odor intensity using a continuous scale ranging from 0 (no attribute) to 10 (extremely intense). Ten grams of koumiss were weighed into a disposable cup for sensory evaluation. To prevent olfactory fatigue, the panel evaluated six samples during a period of 60 min, taking a break after assessing three samples. Three replicates were performed for each sample, and six experiments were conducted, with sensory data collected by trained assessors. To ensure the accuracy and reliability of sensory evaluation, we engaged professional personnel from Alar Xin Nong Dairy Industry Co., Ltd., Alar, China, who specialize in dairy product research and development, to provide guidance and supervision on our sensory analysis work.

### 2.5. Odor Activity Value Calculation

Utilize the methodology of OAV to assess the olfactory contribution of various regional koumiss. OAV refers to the ratio between the mass concentration of an aroma compound and its sensory threshold in an odor system.

### 2.6. Statistical Analysis

The differences in key components among the koumiss products from different regions were explored using partial least squares discriminant analysis (PLS-DA) with SIMCA-P 13.0 software (Umetrics, Umea, Sweden). The differences in volatile compounds and threshold values among different regions were evaluated using one-way analysis of variance (ANOVA) with SPSS Statistics 20.0 software (SPSS Inc., Chicago, IL, USA), with a significance level of *p* < 0.05. Bar graphs, Venn diagrams, and radar charts were produced using Origin 2021 software (OriginLab Corporation, Northampton, MA, USA).

## 3. Results and Analysis

### 3.1. GC-IMS Was Used to Establish the Fingerprint of Volatile Substances in Koumiss from Different Regions

Currently, GC-IMS is a new emerging instrument analysis technique that can effectively separate ions based on their migration speed at atmospheric pressure, eliminating the need for a vacuum environment and increasing convenience. It has the advantages of high sensitivity and data visualization and is widely used for origin tracing and quality control.

We utilized GC-IMS to investigate the volatile fingerprint of traditional fermented koumiss of the Kazakh ethnic group in four regions of Xinjiang. Data was generated using the FlavourSpec^®^ instrument, as shown in Figure 2. Figure 2a presents a top-down view of the 3D spectrum of the volatile components of koumiss. The blue background of the entire figure represents the ion migration time along the X-axis and the retention time of the ions in the gas chromatography on the Y-axis. The red vertical line at 1.0 on the X-axis represents the normalized response ion peak of the volatile components because subtle changes in capillary column temperature and carrier gas flow rate can lead to changes in retention time. The point to the right of the cleavage represents a volatile substance, with the color indicating the signal strength of individual compounds. The higher the concentration of the compound, the redder the color, and conversely, the lower the concentration, the bluer the color. As shown in Figure 2a, most signals are distributed within the retention time range of 100–1200 s, and many peaks have similar signal distributions in the samples. Still, the peak intensities differ significantly between koumiss from different regions, indicating differences in aroma content.

To compare the differences between samples, Figure 2b shows that a differential contrast method can be used. One spectrum from a sample is chosen as the reference sample, and other spectra are subtracted for comparison. If the volatile substances in the two samples are the same, the background after subtraction is white, with red representing higher concentrations than the reference sample, and blue representing lower concentrations. As shown in Figure 2b, when comparing sample YL-2 with reference sample YL-1 and other samples, the blue and red areas are significantly reduced, indicating that the volatile substances of YL-1 and YL-2 are similar in concentration, while there are significant differences in concentration between YL-1 and other samples. By establishing differential contrast maps for reference samples from different regions using GC-IMS, we can preliminarily discriminate and trace the koumiss from different regions.

Additionally, Figure 2c presents the GC-IMS volatile organic compound fingerprint patterns of different samples, which were created by piecing together screenshots of each volatile organic compound from Figure 2a. Each row represents a sample, and each column represents a compound. By drawing the fingerprint patterns, we can clearly compare the volatile organic compound contents and changes in each region’s samples. In total, 35 volatile organic compounds were qualitatively determined. Based on Figure 3 and previous research results [18], the results of volatile organic compounds identified by GC-IMS were considered insufficient. Although the results cannot fully demonstrate the flavor characteristics of koumiss from different regions, they can establish fingerprint patterns for grading and discrimination. By selecting the region outlined by the red box in Figure 2c, it can be observed that the concentrations of dimethyl sulfide, ethyl butyrate, ethyl hexanoate, ethyl octanoate, and 2-methylbutanal in the samples from the TC region decreased significantly with the change of the region. In the region outlined by the yellow box, the concentrations of isobutyl acetate, ethyl propanoate, 2-methylpropanoic acid ethyl ester, propyl acetate, and 1-propanol in the samples from the ALTe region were significantly higher than those in other regions. In the region outlined by the green box, the concentrations of trans-2-pentenal, methyl 2-methylbutyrate, hexanal, (E)-2-hexenal, 1-penten-3-ol, 1-penten-3-one, and acetone in the samples from the ALTw region were significantly higher than those in other regions.

### 3.2. Analysis of Differences in Volatile Compounds of Koumiss from Different Regions Based on PLS-DA Technology

PLS-DA is a supervised discriminant analysis statistical technique. It establishes a relationship model between volatile compounds and sample categories to predict sample categories [19]. Therefore, in this study, a PLS-DA model was adopted to identify specific marker compounds for the volatile substance differences of koumiss samples from four regions based on the detected volatile substance content data matrix.

As shown in Figure 4a, the four groups of samples were successfully distinguished in the quadrant plot. Specifically, the samples from the YL region were in the fourth quadrant, those from the TC region were in the first quadrant, samples from the ALTw region were in the second quadrant, and samples from the ALTe region were in the third quadrant. Additionally, as shown in Figure 4b, 200 permutation tests were performed, and the model predictive index (Q2) was 0.69 for the prediction of different origins of koumiss, indicating a prediction accuracy of 69%. A Q2 value exceeding 0.5 indicates an acceptable fit of the model. The intersection of the Q2 regression line and the vertical axis is less than zero, indicating the absence of overfitting, thus validating the model’s effectiveness [20]. The variable importance in projection (VIP) method was employed to select the most statistically significant variables. It is generally recognized that volatile substances with VIP values >1.0 play a crucial role in distinguishing geographical origins [21]. As shown in Figure 4c, 18 substances were selected based on their VIP values, including 2-heptanone, ethyl butyrate, dimethyl sulfide, 2-methylbutanal, ethyl acetate, isobutyraldehyde, methyl ethyl ketone, isobutyl acetate, isopentyl acetate, acetone, ethyl propionate, nonanal, ethyl caproate, isobutanol, propyl acetate, amyl alcohol, ethyl 2-methylpropanoate, and n-propanol.

To ensure the accuracy of identifying the selected key volatile compounds’ traceability for koumiss from different regions, a standard was adopted that included volatile compounds with VIP values greater than 1 and shared by GC-IMS fingerprinting discrimination of volatile compounds. Based on this standard, the following eight volatile compounds were selected as the final markers for distinguishing koumiss from different regions: dimethyl sulfide, ethyl butyrate, 2-methylbutanal, ethyl propionate, ethyl 2-methylpropanoate, isobutyl acetate, propan-1-ol, and methyl ethyl ketone.

### 3.3. Comparison of GC-MS and GC-IMS Results

Both GC-MS and GC-IMS are excellent instruments for determining volatile compounds, and the research results can better understand the characteristics of the two instruments. The results in Table 2 show that 65 volatile compounds were identified by GC-MS in koumiss samples from four regions. 35 volatile compounds were identified by GC-IMS, of which IMS identified 76 peaks, but 26 peaks were not characterized. Although IMS has a strong sensitivity to the ion drift of volatile compounds without fragmentation and can identify 76 peaks, 26 volatile compounds cannot be confirmed due to incomplete databases. Although GC-IMS is much smaller than GC-MS in the number of substances determined, GC-IMS can conveniently sample and test and quickly obtain results. Among the 50 qualitatively determined volatile compounds, 15 were diploids. When the charged volatile substance is given water and a hydrogen ion, there is a chance that a monomer and two monomers will combine to form a haploid and a diploid [22], making the quantification of volatile compounds by IMS more accurate. Analyzing the types of identified compounds, among the 35 volatile compounds identified by GC-IMS, there are 14 ester compounds, six alcohol compounds, six ketone compounds, and eight aldehyde compounds. Compared with the pattern of identified compounds by GC-MS, GC-IMS is more sensitive to aldehyde and ketone compounds and is more suitable for samples with more aldehyde and ketone compounds. This is consistent with the research results of Zhang [23]. In addition, as shown in Figure 3, 13 common volatile compounds were identified by these two types of analysis methods, and a total of 87 volatile compounds were identified, which is much more abundant than the substances determined by a single detection method. Therefore, the combination of GC-IMS and GC-MS can more comprehensively identify and quantify volatile compounds.

### 3.4. The Characteristic Volatile Compounds of Koumiss from Different Regions Were Analyzed by GC-MS Combined with GC-IMS

To investigate the aroma characteristics of traditional fermented Kazakh koumiss from different regions, a combined GC-IMS and GC-MS method was used to analyze and identify the volatile compounds and their relative contents in samples from four regions in Xinjiang. As shown in Table 2, a total of 87 volatile flavor substances were detected, including 16 kinds of alcohol compounds (excluding ethanol), 33 kinds of ester compounds, 15 kinds of aldehyde compounds, nine kinds of ketone compounds, three kinds of acid compounds, and one kind each of alkane, phenol, and ether compounds. Compared with the report by Wang [24], more volatile compounds were detected using the combined detection method. As shown in Figure 5, the aroma component types of koumiss samples from the four regions were similar, but the total concentrations of each type varied. The highest content of aroma components was ester compounds, followed by acid compounds, alcohol compounds, and ketone compounds. Ester compounds, acid compounds, and alcohol compounds accounted for a large proportion of the total and were important volatile compounds in koumiss.

Sourness is the main component of the flavor of koumiss. The relative content of caprylic acid, capric acid, 9-decenoic acid, caproic acid, and acetic acid is relatively high in koumiss samples from different regions. Among them, medium-chain fatty acids caprylic acid, capric acid, 9-decenoic acid, and caproic acid account for 97% of the total volatile acids, giving koumiss a pungent, sour taste and a faint odor, similar to the results of Du’s research [25]. This may be due to the abundant free fatty acids in mare’s milk or the breakdown of fats by microorganisms. Compared with cereal wines, which are also fermented by lactic acid bacteria and yeast, the fatty acids in koumiss are mostly oleic acid and linoleic acid [26], different from the metabolism and breakdown of fatty acids in fermented dairy products. The acetic acid in koumiss is sour but not astringent, formed by the metabolism of lactose in mare’s milk by some β-galactosidase-producing microorganisms [27]. The total volatile acid content of the YL and ALTe regions is significantly higher than that of the TC and ALTw regions (see Figure 5). Due to the special process of fermenting koumiss in the YL region, the machine stirring method was used in the fermentation process (Table 1 and Figure 1). Compared with hand-stirring fermentation, mechanical stirring fermentation extends the stirring time, so that microorganisms do not settle at the bottom and obtain more oxygen, which can fully ferment and produce volatile acid compounds. However, the TC region also used mechanical stirring to ferment koumiss. Still, the content of volatile acid compounds is significantly lower than that of other regions, which may be related to ester volatile compounds.

The esters of volatile substances are important flavor compounds in koumiss due to their very low threshold. Esters make up an average of 63% of the total content of koumiss, and impart a buttery, fruity, and floral aroma to it, helping to reduce the off-flavors of fatty acids and miscellaneous alcohols in koumiss [28]. Therefore, this class of compounds plays an important role in forming the characteristic aroma of koumiss. Most of the esters in koumiss are produced by two pathways. The first pathway involves the esterification of short-chain and medium-chain fatty acids, free fatty acids, and primary and secondary alcohols produced during lactose fermentation or amino acid metabolism in koumiss, and the second pathway involves the formation of esters by alcohol acyltransferases produced by microorganisms from alcohols and acyl-CoA [29,30]. As shown in Figure 5, the total content of esters of volatile substances in the YL and TC regions is significantly higher than that in the ALT region, which may be due to the diversity of fermentation agents and the special nature of the process used in the fermentation of koumiss in these regions. The activity and enzymatic activity of yeast in the YL and TC regions are enhanced by the climate and fermentation methods used there, which can fully promote the formation of esters. The esters with relatively high relative contents are ethyl caprylate, ethyl octanoate, ethyl laurate, and ethyl 9-decenoate, which are positively correlated with the types and contents of volatile acids. Yu’s [31] research also showed that ethyl octanoate and ethyl caprate are important components of esters of volatile substances in koumiss.

Alcohols with volatile substances are one of the main flavor compounds differentiating koumiss from different regions. Most high-level alcohols with high content in koumiss include phenylethanol, isopentanol, and isobutanol. Phenylethanol is a widely used edible fragrance with a rose scent, usually employed as a flavoring agent for various food preparations. Dairy products contain a rich amount of phenylalanine. In the fermentation process of koumiss, the majority of phenylethanol is generated by the transamination and decarboxylation metabolism of phenylalanine via the Ehrlich reaction by yeast [32]. The content of phenylethanol in the ALTe region is as high as 3568.9 μg/kg, significantly higher than the 941 μg/kg in the YL region, possibly due to the different fermentation environments and fermentation strains. Phenylethanol, isopentanol, and isobutanol are common fusel oils in brewing, and the content of fusel oils determines some aroma characteristics and taste of koumiss. Excessively high content can cause severe headaches and pose a hazard to human health [33]. Similar to phenylethanol, most of the isopentanol and isobutanol are produced through the Ehrlich reaction with leucine and valine as precursors.

Table 2 shows that aldehydes and ketones are the main volatile compounds that differentiate the flavors of koumiss in different regions. Among the high-content aldehydes in koumiss, hexanal, trans-2-octenal, and acetaldehyde are the most prominent. Aldehydes in the fermentation process mainly come from the metabolism of fatty acids, the transamination of amino acids, or the Strecker degradation [34]. However, aldehydes are unstable in koumiss and can be easily reduced to the corresponding alcohols and fatty acids under certain conditions, resulting in relatively low relative content [35]. Trans-2-octenal usually has a fatty and meaty aroma. It was found to be highest in samples from the ALTw region and was not detected in samples from the TC and YL regions. The content of ketones in koumiss is relatively low, mainly consisting of acetoin and acetone. Acetoin is a metabolic product of some microorganisms that balance the acid-base level and store energy, produced by the glycolytic pathway of pyruvate [36]. The highest content of acetoin was found in the ALTw region, which was significantly higher than in other regions.

In summary, it can be speculated that geographical environment, fermentation temperature and method, and the type of fermentation agent may be important factors that influence the differences in the aroma of traditional fermented Kazakh koumiss in different regions.

### 3.5. Analysis of OVA in Koumiss from Different Regions

Different regions of koumiss have unique flavors, mainly determined by the types, amounts, and corresponding thresholds of volatile flavor substances they contain. Studies have shown that the contribution of a single aroma component to the overall aroma of koumiss depends on its concentration and odor threshold [37]. Although the above analysis provides a quantitative characterization of the flavor substances in koumiss, it is only by understanding the threshold of each substance that the key flavor substances can be directly linked to their contribution to the overall aroma of koumiss. The OAV can be used to evaluate the composite contribution of flavor substances to food. Compounds with an OAV ≥1 are generally considered to be aroma-contributing components in koumiss, while substances with an OAV ≥10 are important aroma-contributing components [38]. The results (shown in Table 3) indicate that 20 volatile flavor substances can be used to calculate OAV, and their OAV values are all greater than 1. Among them, the OAV values of caprylic acid ethyl ester, caprylic acid, 9-decenyl acetate, decanoic acid ethyl ester, decanoic acid, isoamyl alcohol, phenethyl alcohol, and hexanoic acid ethyl ester in koumiss from different regions are all higher than 10, indicating that these volatile substances may be key flavor components in koumiss.

The results are shown in Table 3. The YL region stands out from other regions regarding its OVA values, with notable components including ethyl decanoate, ethyl octanoate, and ethyl 9-decenoate. In particular, ethyl decanoate and ethyl octanoate in the YL region have OVA values more than seven times higher than those in the ALT region and are therefore the most important characteristic aromas of the YL region. Ethyl decanoate and ethyl octanoate impart fruity and creamy aromas and have OVA values exceeding 100 in all regions of koumiss and have repeatedly been confirmed as key aroma components in koumiss from different regions of China [41]. The OVA value of phenethyl alcohol in the ALTe region is significantly higher than in other areas, making it one of the main aroma components that affect koumiss in this region. It is worth noting that previous studies on koumiss in other regions did not find phenethyl alcohol as a leading aroma sample, making it a unique flavor in the traditional fermented koumiss of the Kazakh ethnic group in Xinjiang. Both octanoic acid and decanoic acid, as sour aromas with OVA values greater than 10, contribute positively to the koumiss flavor and the dairy aroma of koumiss, providing a unique and distinctive taste.

### 3.6. Sensory Analysis of Koumiss in Different Regions

As the aroma components of koumiss samples measured by instruments cannot well explain the overall flavor of koumiss, and the aroma components contained in koumiss cannot explain their interactions, the human olfactory sensory system can provide comprehensive and sensitive feedback on the sensory characteristics of koumiss. Quantitative descriptive analysis (QDA) can obtain a complete sensory description of the sample, thus clearly showing the differences between samples [42]. To quantitatively describe the aroma of koumiss, it is crucial to develop sensory vocabulary with definitions and scales. Our sensory research team eventually determined the sensory evaluation vocabulary of fruit, cream, floral, odor, milk, and sourness. A radar chart was drawn based on the obtained sensory analysis data, as shown in Figure 6. For the attributes of fruit and cream, the scores of YL and TC regions were significantly higher than those of ALTe and ALTw regions, which were the main attributes of YL and TC regions. This may be due to the high content of ethyl caprylate and decanoic acid ethyl ester in YL and TC regions, consistent with the prediction of OVA values. The odor attribute had only a relatively high intensity in the YL region, which may be a manifestation of excessively high levels of Octanoic acid and n-decanoic acid. For the milky attribute, the ALTw region had the highest score, with an intensity of 4.2. The ALTe region had the highest score for the floral attribute, which was related to the content of phenylethanol. Interestingly, the sensory profile characteristics of koumiss in different regions are quite different. These phenomena indicate that the metabolic capacity of flavor substances produced in the fermentation process of koumiss differs due to different fermentation conditions and other factors, which provides a theoretical basis for establishing and differentiating koumiss fermentation techniques with specific sensory characteristics.

Referring to Liu’s classification method for fermented dairy products [13], the following rules were applied for aroma classification: if a specific aroma attribute of a fermented dairy sample scored the highest, it was categorized under that aroma type. Therefore, the four different regions of koumiss were classified into three aroma types, with the TC and YL regions falling under the fruit-cream type, ALTw under the milk type, and ALTe under the floral type. Currently, there is no other research on the aroma types of koumiss and the aroma classification results of this study can provide suggestions for establishing aroma classification of koumiss in the future.

## 4. Conclusions

Using gas chromatography coupled with ion mobility spectrometry, we established fingerprint profiles of koumiss from four different regions. Combination of the application of PLS-DA analysis, we identified dimethyl sulfide, ethyl butyrate, 2-methylbutanal, ethyl propionate, 2-methylpropyl propionate, isobutyl acetate, n-propanol, and methyl ethyl ketone as indicative markers for the rapid detection of traditionally fermented koumiss from these regions. In conclusion, it is worth emphasizing that the use of GC-IMS makes it possible to identify regions of origin, and thus control the authenticity of food origin. This is important for different food groups but for regional and traditional food. By utilizing GC-IMS and GC-MS to analyze the volatile components of the koumiss from these regions, we found that the volatile components were similar in terms of variety. However, due to differences in the fermentation process between regions, the volatile components exhibited significant differences in concentration and displayed clear regional characteristics. A total of 87 volatile components were detected, with esters, acids, and alcohols being the main aromatic substances in the koumiss. Furthermore, comparing these two detection methods regarding volatile compound analysis revealed that GC-IMS exhibited a more sensitive response towards aldehydes and ketones. Subsequently, an analysis of the OAV for koumiss from different regions identified ethyl caprylate, ethyl caprate, octanoic acid, 9-decenoic acid ethyl ester, n-decanoic acid, isopentanol, phenethyl alcohol, and ethyl hexanoate as key aroma components in traditional fermented koumiss of the Kazakh ethnic group. Interestingly, the aroma profile of the ALTw region was primarily dominated by phenethyl alcohol, while the YL and TC regions showcased a prominent presence of ethyl caprylate and ethyl caprate. Additionally, a qualitative descriptive analysis (QDA) was conducted to evaluate and categorize the aroma profiles of koumiss from the four regions, resulting in three distinct aroma types: fruit-cream type, milk type, and floral type.

## Figures and Tables

**Figure 1 foods-12-02223-f001:**
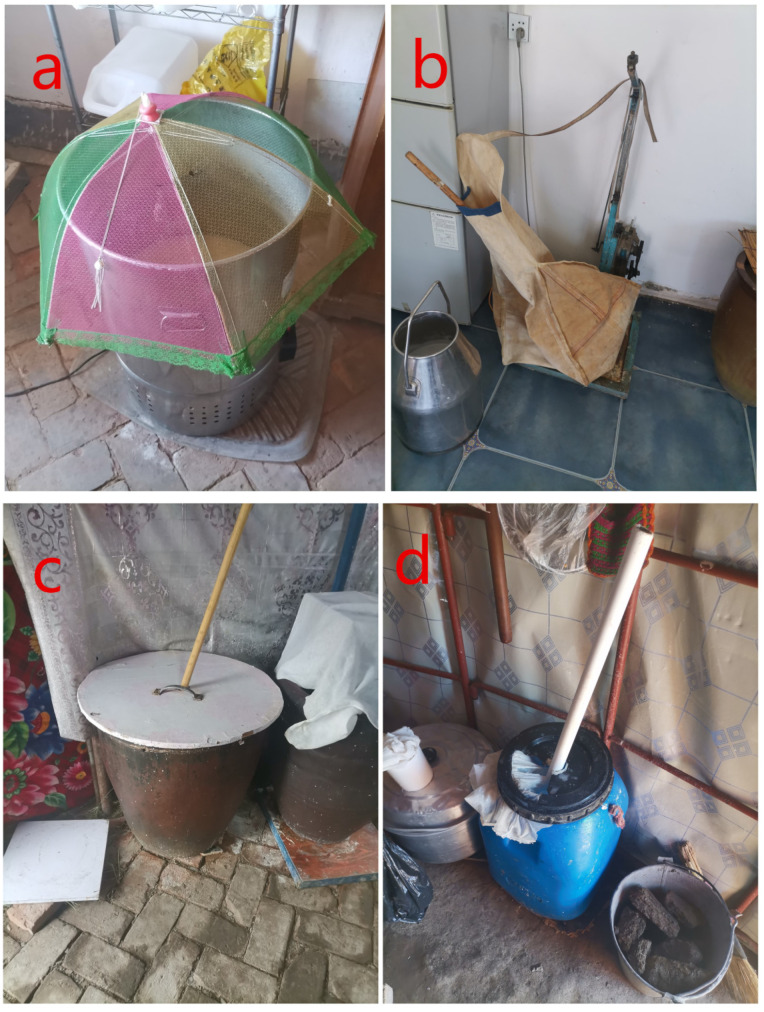
Depicts the fermentation methods of traditional koumiss from different regions. Panel (**a**) represents the YL region, panel (**b**) the TC region, panel (**c**) the ALTe region, and panel (**d**) the ALTw region.

**Figure 2 foods-12-02223-f002:**
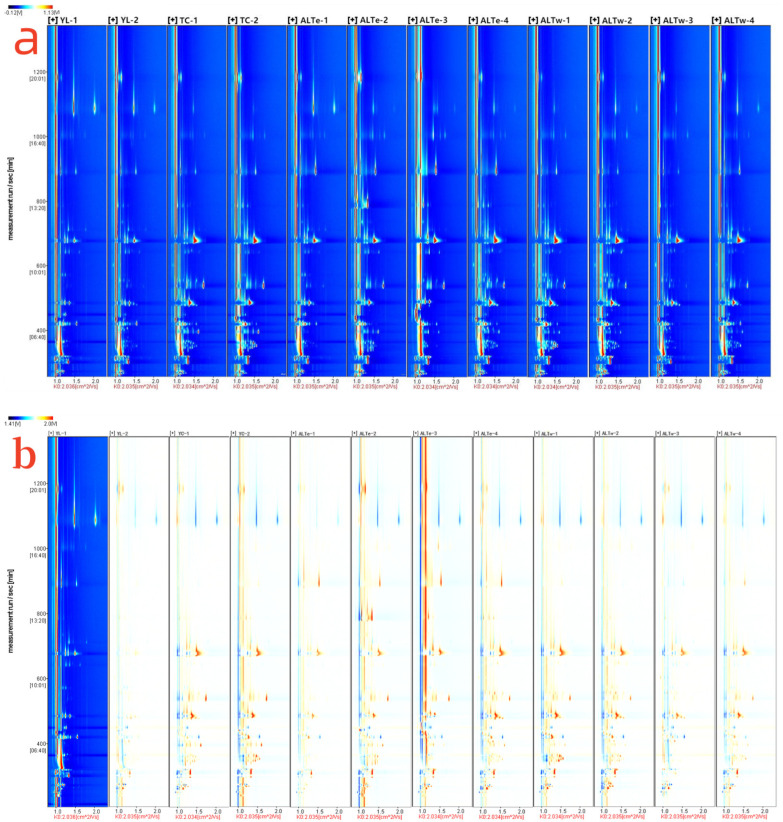
GC-IMS analysis of koumiss from different regions. (**a**) Top view of the 3D spectrum of the volatile components of koumiss from different regions. (**b**) 2D comparison chart of the volatile components of koumiss from different regions. (**c**) The fingerprint spectra of the volatile constituents in koumiss from distinct regions. The delineated color zones represent the variances in volatile substances among different regions.

**Figure 3 foods-12-02223-f003:**
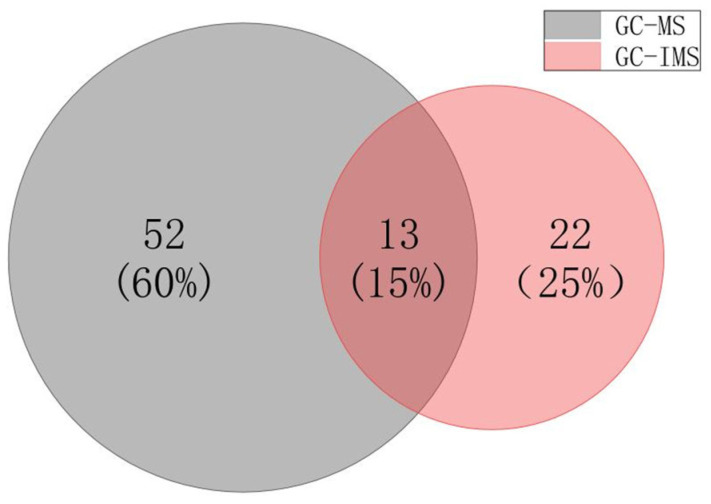
The Wayne diagram of the number of aroma substances determined by GC-MS and GC-IMS.

**Figure 4 foods-12-02223-f004:**
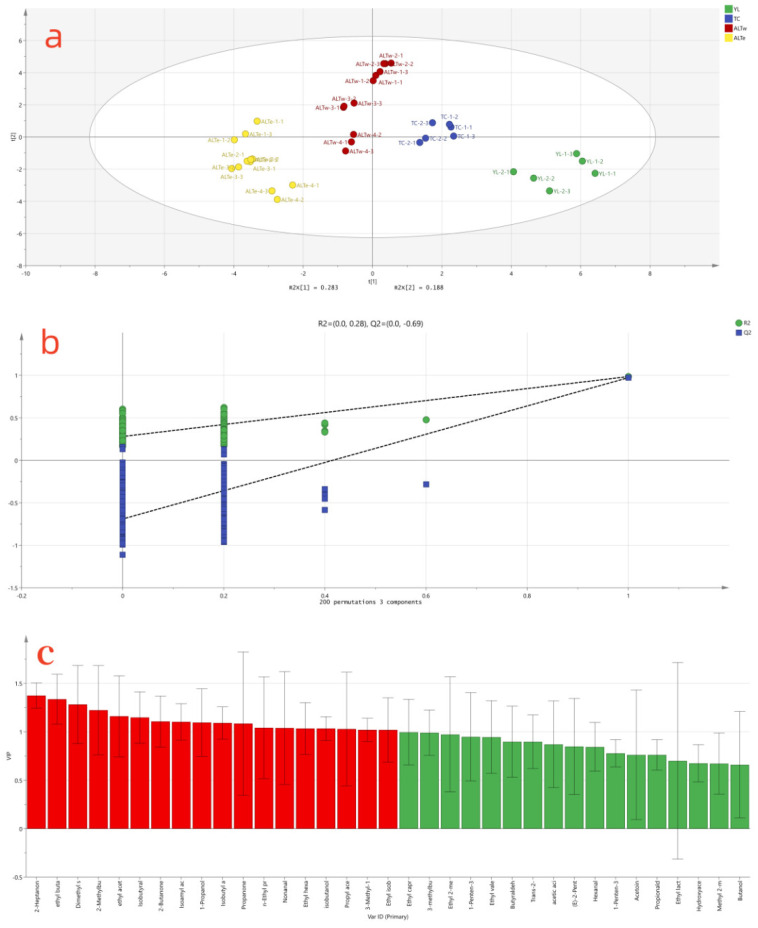
PLS-DA analysis of koumiss from different regions. (**a**) Score plot of PLS-DA; (**b**) Cross-validation results based on 200 permutation tests; (**c**) Ranking of volatile substances based on VIP scores. The red part is the substance with VIP value greater than 1. The green area is for substances with VIP finger less than 1.

**Figure 5 foods-12-02223-f005:**
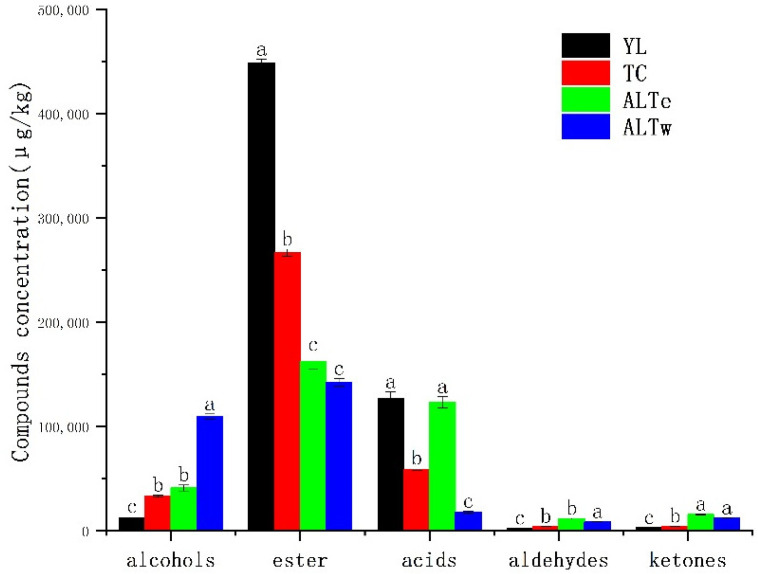
Changes in the total content of alcohols, acids, esters, aldehydes, and ketones in koumiss from different regions. Letters a, b, and c indicate significant differences (*p* > 0.05) in the content of these compounds among different samples.

**Figure 6 foods-12-02223-f006:**
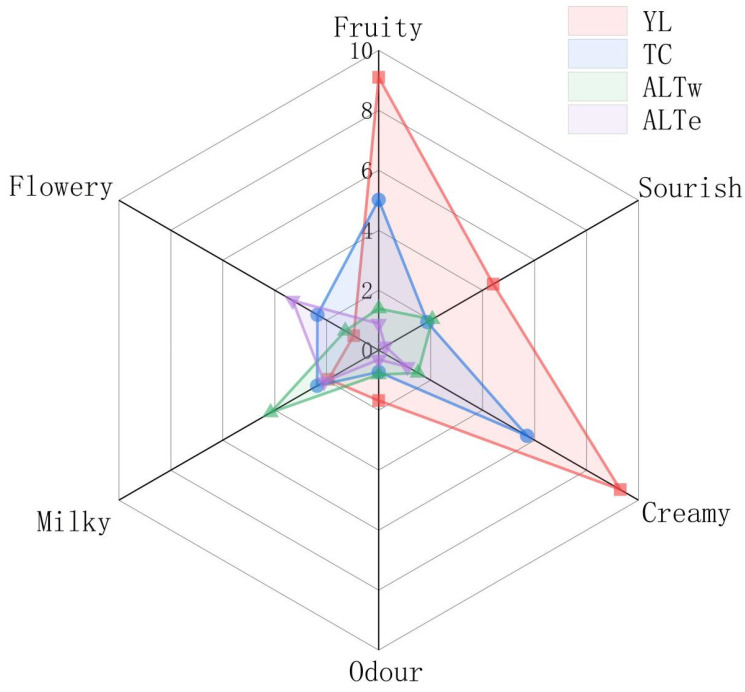
Radar chart of sensory analysis of koumiss from different regions.

**Table 1 foods-12-02223-t001:** Koumiss sample information.

Abbreviation of Koumiss	Locality	Fermentation Method	Fermentation Temperature °C
YL	Ili	machinery	20~26
TC	Tacheng	machinery	22~26
ALTe	Altai east	manually	22~27
ALTw	Altai west	manually	21~27

**Table 2 foods-12-02223-t002:** Volatile substance content of koumiss in different regions.

No.	Cas	VOCs	Identification	Contents (μg/kg)			
				YL	TC	ALTw	ALTe
C1	60-12-8	Phenethyl alcohol	MS	941.79 ± 640.57 c	2483.21 ± 442.29 ab	1373.66 ± 1395.45 bc	3568.09 ± 1500.03 a
C2	78-83-1	isobutanol	MS, IMS e	47.98 ± 31.67 c	158.28 ± 55.17 b	122.32 ± 131.17 bc	403.57 ± 103.51 a
C3	24347-58-8	(2R,3R)-(−)-2,3-Butanediol	MS	43.97 ± 29.01 a	61.04 ± 2.46 a	47.33 ± 38.37 a	52.97 ± 12.09 a
C4	123-51-3	3-Methyl-1-butanol	MS, IMS e	552.43 ± 314.88 c	1761.30 ± 372.16 b	1110.76 ± 944.97 bc	3464.17 ± 717.10 a
C5	1565-80-6	(S)-(−)-2-Methylbutanol	MS	183.61 ± 201.13 a	0	178.36 ± 341.27 a	295.30 ± 534.22 a
C6	67-63-0	Isopropanol	MS	44.14 ± 74.77 a	18.82 ± 6.38 a	112.60 ± 163.46 a	23.51 ± 34.63 a
C7	50551-88-7	5-METHYL-5-HEXEN-2-OL	MS	7.27 ± 8.60 a	18.63 ± 12.22 a	19.76 ± 41.02 a	10.66 ± 23.26 a
C8	137-32-6	2-Methylbutan-1-ol	MS	177.12 ± 194.24 b	876.01 ± 158.21 a	237.95 ± 322.88 b	1017.48 ± 282.73 a
C9	2313-61-3	4-methyl-2-hexanol	MS	30.83 ± 33.99 a	26.65 ± 1.61 a	61.71 ± 112.77 a	0
C10	19132-06-0	(2S,3S)-(+)-2,3-BUTANEDIOL	MS	0	80.78 ± 25.91 a	21.46 ± 39.04 a	124.81 ± 250.74 a
C11	108-82-7	2,6-Dimethyl-4-heptanol	MS	0	0	82.38 ± 84.49 a	117.19 ± 137.57 a
C12	71-36-3	Butanol	MS	5.04 ± 0.52 a	2.87 ± 0.21 a	5.22 ± 3.58 a	4.39 ± 0.62 a
C13	616-25-1	1-Penten-3-ol	MS	1.23 ± 0.26a	3.79 ± 1.26 a	2.93 ± 3.29 a	3.61 ± 3.28 a
C14	71-23-8	1-Propanol	MS	46.37 ± 10.19 b	42.89 ± 14.84 b	43.83 ± 15.33 b	52.61 ± 23.54 a
C15	763-32-6	3-methylbut-3-en-1-ol	MS	0.66 ± 0.18 b	0.83 ± 0.16 ab	0.76 ± 0.35 ab	1.01 ± 0.32 a
C16	112-60-7	2,2′-((Oxybis(ethane-2,1-diyl))bis(oxy))diethanol	MS	647.70 ± 764.98 a	858.08 ± 220.96 a	762.76 ± 940.57 a	0
Z1	110-38-3	Ethyl caprate	MS	28,937.46 ± 2643.22 a	14,466.14 ± 1822.56 b	4222.47 ± 2435.52 c	2445.60 ± 1629.19 c
Z2	106-32-1	Ethyl caprylate	MS, IMS e	24,254.33 ± 2941.10 a	15,167.06 ± 2536.12 b	3865.41 ± 1682.78 c	2890.76 ± 1198.86 c
Z3	106-33-2	Ethyl laurate	MS	7819.93 ± 1002.54 a	5144.83 ± 376.36 b	589.22 ± 323.47 c	274.81 ± 278.27 c
Z4	50816-18-7	9-Decen-1-ol, 1-acetate	MS	8430.05 ± 753.10 a	3831.73 ± 279.66 b	1707.36 ± 1320.09 c	941.79 ± 906.60 c
Z5	687-47-8	(−)-Ethyl L-lactate	MS	1877.55 ± 853.21 a	1009.89 ± 606.72 b	461.19 ± 366.60 c	1129.09 ± 357.94 b
Z6	124-06-1	Ethyl tetradecanoate	MS	626.97 ± 99.88 a	467.33 ± 74.17 b	18.30 ± 14.13 c	14.60 ± 18.24 c
Z7	123-29-5	Ethyl nonanoate	MS	445.54 ± 192.25 a	0	260.64 ± 474.45 a	0
Z8	123-66-0	Ethyl hexanoate	MS, IMS	306.92 ± 22.82 a	263.36 ± 44.09 b	46.65 ± 28.70 c	62.93 ± 14.65 c
Z9	141-78-6	ethyl acetate	MS, IMS	651.07 ± 446.58 b	1265.80 ± 450.34 ab	1177.51 ± 1129.16 b	2071.71 ± 693.74 a
Z10	54546-22-4	9-Hexadecenoic acid,ethyl ester	MS	167.27 ± 29.15 a	94.63 ± 37.19 b	0	0
Z11	2035-99-6	3-methylbutyl octanoate	MS	126.86 ± 39.42 a	70.28 ± 23.46 b	0	0
Z12	628-97-7	Ethyl palmitate	MS	73.81 ± 20.29 a	55.57 ± 5.10 b	0	0
Z13	105-54-4	ethyl butanoate	MS, IMS	59.25 ± 29.14 b	135.77 ± 22.49 a	2.45 ± 4.43 c	7.66 ± 8.12 c
Z14	10348-47-7	ethyl 2-hydroxy-4-methyl valerate	MS	121.85 ± 53.43 b	538.80 ± 119.68 a	113.12 ± 113.58 b	68.03 ± 23.09 b
Z15	624-13-5	propyl octanoate	MS	69.30 ± 17.51 a	60.29 ± 5.17 a	8.21 ± 10.02 b	9.15 ± 6.10 b
Z16	106-30-9	Ethyl heptanoate	MS	46.49 ± 6.88 a	32.76 ± 2.65 a	13.89 ± 25.46 b	0
Z17	2277-23-8	1-Decanoyl-rac-glycerol	MS	16.81 ± 7.78 a	12.29 ± 13.46 ab	0	12.78 ± 20.58 ab
Z18	110-19-0	Isobutyl acetate	MS, IMS	157.60 ± 82.56 a	59.80 ± 9.95 b	0	8.38 ± 9.42 c
Z19	142-91-6	Isopropyl palmitate	MS	9.62 ± 3.12 a	68.20 ± 97.21 a	0	0
Z20	547-64-8	1-(Dibromomethyl)-3-phenoxybenzene	MS	198.66 ± 222.76 a	0	139.81 ± 111.72 ab	28.10 ± 55.04 b
Z21	97-64-3	Ethyl lactate	MS, IMS	49.63 ± 55.55 b	368.21 ± 182.02 a	52.18 ± 61.10 b	96.82 ± 122.33 b
Z22	15399-05-0	ethyl phenyl lactate	MS	30.12 ± 33.53 b	104.55 ± 17.42 a	6.42 ± 11.65 c	0
Z23	55130-16-0	benzyl oleate	MS	27.34 ± 30.21 bc	43.65 ± 13.95 ab	74.60 ± 60.89 a	1.46 ± 2.64 c
Z24	103-45-7	Phenethyl acetate	MS	0	733.98 ± 95.20 ab	320.60 ± 584.85 b	1088.80 ± 583.44 a
Z25	123-92-2	Isoamyl acetate	MS, IMS	0	44.12 ± 49.27 b	68.29 ± 67.81 b	130.11 ± 46.85 a
Z26	19329-89-6	lactic acid isoamyl ester	MS	0	53.88 ± 59.30 ab	36.60 ± 39.34 ab	80.05 ± 70.30 a
Z27	624-41-9	2-Methylbutylacetat	MS	0	49.68 ± 21.70 a	11.91 ± 22.99 b	26.15 ± 36.63 ab
Z28	539-82-2	Ethyl valerate	IMS	129.20 ± 16.61 a	66.45 ± 10.05 b	85.55 ± 42.10 b	99.60 ± 47.47 ab
Z29	7452-79-1	Ethyl 2-methylbutanoate	IMS	58.05 ± 23.25 a	16.78 ± 3.14 b	8.69 ± 3.06 ab	8.00 ± 4.90 b
Z30	97-62-1	Ethyl isobutyrate	IMS	15.72 ± 1.84 ab	97.88 ± 51.61 ab	58.80 ± 72.58 b	118.66 ± 118.51 a
Z31	868-57-5	Methyl 2-methylbutyrate	IMS	11.66 ± 0.56 c	51.97 ± 16.46 a	30.34 ± 17.33 b	47.28 ± 16.41 a
Z32	105-37-3	n-Ethyl propanoate	IMS	10.18 ± 2.16 b	41.22 ± 12.76 b	95.06 ± 89.10 ab	184.14 ± 202.16 a
Z33	109-60-4	Propyl acetate	IMS	4.07 ± 1.26 a	9.09 ± 6.11 a	30.12 ± 31.96 a	24.58 ± 26.11 a
S1	124-07-2	Octanoic acid	MS	11,231.01 ± 3510.55 a	4556.99 ± 720.53 b	5045.42 ± 3687.15 b	747.79 ± 617.20 c
S2	334-48-5	Decanoic acid	MS	5546.54 ± 1855.71 a	1946.03 ± 309.87 b	2062.60 ± 1556.39 b	158.60 ± 302.53 c
S3	14436-32-9	9-Decenoic acid	MS	1459.96 ± 407.24 a	294.40 ± 322.50 b	411.40 ± 302.25 b	0
S4	142-62-1	1-Hexanoic acid	MS	1043.66 ± 53.25 a	409.12 ± 107.38 b	278.01 ± 138.91 c	81.90 ± 87.16 d
S5	64-19-7	acetic acid	MS, IMS	472.64 ± 88.24 bc	726.91 ± 141.36 ab	1045.17 ± 621.60 a	196.39 ± 144.50 c
S6	143-07-7	Lauric acid	MS	464.89 ± 142.63 a	205.86 ± 297.01 b	18.22 ± 32.95 c	0
S7	3004-93-1	S-2-methyl octanoic acid	MS	49.38 ± 38.43 a	114.60 ± 45.29 a	0	0
S8	503-74-2	3-Methylbutanoic acid	MS	71.11 ± 55.90 a	64.17 ± 15.38 a	5.47 ± 9.95 b	51.20 ± 7.75 a
S9	65-85-0	benzoic acid	MS	148.38 ± 162.54 c	481.30 ± 528.87 b	572.45 ± 375.91 a	0
S10	629-56-1	Hexadecenoic acid	MS	647.70 ± 764.98 a	858.08 ± 220.96 a	762.76 ± 940.57 a	0
S11	79-31-2	Isobutyric acid	MS	33.76 ± 37.10 b	91.65 ± 29.33 a	27.73 ± 32.63 b	127.28 ± 82.73 a
S12	116-53-0	2-Methylbutyric acid	MS	28.25 ± 31.72 c	73.02 ± 4.55 b	39.12 ± 25.58 c	120.02 ± 42.69 a
T1	110-43-0	2-Heptanone	MS, IMS	30.92 ± 34.20 a	22.89 ± 25.07 a	42.90 ± 47.54 a	24.16 ± 26.96 a
T2	105-42-0	4-Methyl-2-hexanone	MS	20.09 ± 22.06 b	17.42 ± 19.09 b	0	44.40 ± 8.02 a
T3	19093-20-0	(E)-5-octenone	MS	0	0	40.32 ± 40.37 a	16.37 ± 22.08 a
T4	821-55-6	2-Nonanone	MS	0	0	76.28 ± 72.95 a	11.45 ± 7.02 b
T5	67-64-1	Propanone	IMS	212.78 ± 38.55 b	332.06 ± 64.46 a	265.48 ± 70.82 b	326.51 ± 16.94 a
T6	513-86-0	Acetoin	IMS e	144.37 ± 8.56 b	229.70 ± 60.35 ab	484.79 ± 490.56 a	242.56 ± 102.04 ab
T7	78-93-3	2-Butanone	IMS e	26.54 ± 16.34 a	16.72 ± 10.94 ab	10.52 ± 11.76 b	7.14 ± 2.92 b
T8	1629-58-9	1-Penten-3-one	IMS e	6.28 ± 0.67 b	56.96 ± 35.47 a	33.87 ± 34.63 ab	45.78 ± 41.89 a
T9	116-09-6	Hydroxyacetone	IMS	13.12 ± 1.31 a	5.36 ± 0.51 b	8.25 ± 6.46 ab	8.32 ± 4.51 ab
F1	100-52-7	Benzaldehyde	MS	66.50 ± 5.41 a	29.28 ± 6.05 b	55.76 ± 8.12 a	25.35 ± 12.53 b
F2	4313-03-5	(E,E)-2,4-Heptadienal	MS	86.63 ± 37.43 a	93.38 ± 63.69 a	187.31 ± 161.50 a	61.91 ± 112.06 a
F3	66-25-1	Hexanal	MS, IMS	113.35 ± 88.89 a	87.66 ± 27.18 a	150.26 ± 73.26 a	115.66 ± 56.13 a
F4	4748-78-1	4-Ethylbenzaldehyde	MS	11.98 ± 13.20 b	8.16 ± 8.94 c	39.99 ± 6.21 ab	47.62 ± 36.29 a
F5	142-83-6	2,4-Hexadienal	MS	30.91 ± 33.91 a	30.37 ± 33.27 a	57.15 ± 25.37 a	42.66 ± 49.06 a
F6	74094-61-4	octanal propylene glycol acetal	MS	7.93 ± 8.68 b	14.55 ± 16.08 b	126.23 ± 141.30 a	27.61 ± 10.45 b
F7	124-19-6	Nonanal	MS, IMS	0	58.98 ± 13.51 a	94.43 ± 74.07 a	55.89 ± 42.23 a
F8	2548-87-0	(2E)-2-Octenal	MS	0	0	194.19 ± 149.75 a	64.14 ± 63.23 b
F9	3913-81-3	(2E)-2-Decenal	MS	0	0	56.02 ± 39.01 a	42.09 ± 16.90 a
F10	123-38-6	Propionaldehyde	IMS e	57.03 ± 11.14 b	153.87 ± 11.72 a	89.36 ± 38.27 b	161.55 ± 52.85 a
F11	123-72-8	Butyraldehyde	IMS	78.67 ± 6.57 ab	104.61 ± 53.51 a	48.95 ± 26.14 b	69.10 ± 32.39 b
F12	6728-26-3	Trans-2-Hexenal	IMS	6.96 ± 1.40 ab	8.14 ± 1.98 a	4.34 ± 1.88 c	5.66 ± 2.74b c
F13	96-17-3	2-Methylbutanal	IMS	17.35 ± 11.30 a	15.85 ± 8.66 a	30.62 ± 38.90 a	31.57 ± 20.05 a
F14	1576-87-0	(E)-2-Pentenal	IMS e	22.81 ± 2.74 b	92.04 ± 24.48 ab	84.41 ± 40.15 b	171.79 ± 129.48 a
F15	78-84-2	Isobutyraldehyde	IMS	10.56 ± 0.67 c	38.58 ± 26.62 bc	73.35 ± 66.59 ab	94.48 ± 21.60 a
Q1	106-44-5	p-Cresol	MS	0	12.57 ± 13.80 b	13.36 ± 18.02 b	56.41 ± 57.52 a
Q2	629-59-4	Tetradecane	MS	0	0	26.48 ± 34.26 a	20.26 ± 15.90 a
Q3	75-18-3	Dimethyl sulfide	IMS	46.42 ± 25.96 a	48.77 ± 42.66 a	5.89 ± 3.98 b	9.63 ± 4.45 b

Note: The letters a, b, c, and d represent significant differences (*p* < 0.05) in the volatile compounds of koumiss from different regions. The letter e indicates that the compound was detected as a dimer in GC-IMS analysis.

**Table 3 foods-12-02223-t003:** OVA content of volatile substances in koumiss from different regions.

VOCs	Cas	Threshold	OVA			
			YL	TC	ALTw	ALTe
Ethyl caprate	110-38-3	0.02 e	1446.87 ± 120.65 a	723.31 ± 83.19 b	221.12 ± 16.59 c	112.28 ± 77.99 c
Ethyl caprylate	106-32-1	0.022 g	1102.47 ± 122.04 a	689.41 ± 105.23 b	175.70 ± 73.23 c	131.40 ± 52.17 c
Ethyl laurate	106-33-2	3.5 f	2.23 ± 0.26 a	1.47 ± 0.09 b	0.17 ± 0.09 c	0.08 ± 0.07 c
Octanoic acid	124-07-2	0.5 e	22.46 ± 6.41 a	9.11 ± 1.32 b	10.09 ± 7.06 b	1.50 ± 1.18 c
9-Decen-1-ol, 1-acetate	50816-18-7	0.1 g	84.30 ± 6.87 a	38.32 ± 2.55 b	17.07 ± 12.64 c	9.42 ± 8.68 c
Decanoic acid	334-48-5	0.5 e	11.09 ± 3.39 a	3.89 ± 0.57 b	4.13 ± 2.98 b	0.32 ± 0.06 c
1-Hexanoic acid	142-62-1	0.2 e	5.22 ± 0.24 a	2.05 ± 0.49 b	1.39 ± 0.66 b	0.41 ± 0.04 c
Ethyl tetradecanoate	124-06-1	0.5 e	1.25 ± 0.18 a	0.93 ± 0.14 b	0.04 ± 0.03 c	0.03 ± 0.03 c
acetic acid	64-19-7	1.2 e	0.39 ± 0.07 bc	0.61 ± 0.11 ab	1.87 ± 0.50 a	0.16 ± 0.12 c
Phenethyl alcohol	60-12-8	0.045 e	20.93 ± 12.99 b	55.18 ± 8.97 ab	30.53 ± 20.69 b	79.29 ± 31.91 a
Ethyl hexanoate	123-66-0	0.005 f	61.38 ± 4.17 a	52.67 ± 8.05 b	9.33 ± 5.50 c	12.59 ± 2.81 c
3-Methyl-1-butanol	123-51-3	0.25 e	2.21 ± 1.15 c	7.05 ± 1.36 b	4.44 ± 3.62 bc	13.86 ± 2.75 a
3-methylbutyl octanoate	2035-99-6	0.125 f	1.01 ± 0.29 a	0.56 ± 0.17 b	0	0
ethyl butanoate	105-54-4	0.025 f	2.37 ± 1.06 b	5.43 ± 0.82 a	0.1 ± 0.17 c	0.31 ± 0.31 c
(E,E)-2,4-Heptadienal	4313-03-5	0.1 f	0.87 ± 0.34 a	0.93 ± 0.58 a	1.87 ± 1.55 a	0.62 ± 1.07 a
Hexanal	66-25-1	0.005 f	2.27 ± 1.62 a	1.75 ± 0.50 a	3.01 ± 1.40 a	2.31 ± 1.07 a
benzoic acid	65-85-0	0.5 e	0.30 ± 0.30 b	0.96 ± 0.27 ab	1.14 ± 0.72 a	0
Acetoin	513-86-0	0.04 e	3.61 ± 0.20 b	5.74 ± 1.38 b	12.12 ± 11.74 a	6.06 ± 2.44 b
Ethyl valerate	539-82-2	0.025 f	5.17 ± 0.61 a	2.66 ± 0.37 b	3.42 ± 1.61 ab	3.98 ± 1.82 ab
Ethyl 2-methylbutanoate	7452-79-1	0.018 f	3.22 ± 1.18 a	0.93 ± 0.16 b	0.48 ± 0.16 b	0.44 ± 0.26 b

Note: The significant differences (*p* < 0.05) in the OVA values of different volatile compounds in koumiss from various regions are represented by letters a, b, c, and d. The letter e refers to the threshold value as per reference [16]. The letter f refers to the threshold value as per reference [39] with milk as the solvent. The letter g refers to the threshold value as per reference [40].

## Data Availability

All related data and methods are presented in this paper. Additional inquiries should be addressed to the corresponding author.

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
