# Peer review of "Exploring the Characteristic Aroma Components of Traditional Fermented Koumiss of Kazakh Ethnicity in Different Regions of Xinjiang by Combining Modern Instrumental Detection Technology with Multivariate Statistical Analysis Methods for Odor Activity Value and Sensory Analysis"

_foods, 2023, doi:10.3390/foods12112223_

Round 1

Reviewer 1 Report

I really liked the material presented. It has been carefully described and the results obtained have been properly analyzed.

However, it is worth improving the conclusion, which is more of a summary. In conclusion, it is worth emphasizing that the use of gas chromatography coupled with ion mobility spectrometry makes it possible to identify regions of origin, and thus control the authenticity of food origin. This is important for different food groups, but for regional and traditional food in particular

A few detailed comments

Figure 1 a and b - improve quality

Line 339 Cut the text and insert the caption of table No. 3

Suggestions for the future:

It seems that it would be very interesting to analyze the composition of koumiss produced in individual regions and coming from several producers. It would then be possible to answer the question of whether it is possible to determine the diversity of kumiss within one region?

Author Response

Dear Dr

Thank you for the reviewers' comments concerning our manuscript entitled "Exploring the Characteristic Aroma Components of Traditional Fermented koumiss of Kazakh Ethnicity in Different Regions of Xinjiang by Combining Modern Instrumental Detection Technology with Multivariate Statistical Analysis Methods for Odor Activity Value and Sensory Analysis" (ID: 2403182). Those comments are all valuable and very helpful for revising and improving our paper, as well as the important guiding significance to our researches. We have studied comments carefully and have made changes which we hope meet with approval. Revised portion are marked in red in the attachment.

Here is a point-by-point response to the comments and concerns of the reviewers.

1.Comment:

I really liked the material presented. It has been carefully described and the results obtained have been properly analyzed.

Response:

We gratefully appreciate for your insightful comments. Thank you for your appreciation of our experiment and article.

2.Comment:

However, it is worth improving the conclusion, which is more of a summary. In conclusion, it is worth emphasizing that the use of gas chromatography coupled with ion mobility spectrometry makes it possible to identify regions of origin, and thus control the authenticity of food origin. This is important for different food groups, but for regional and traditional food in particular

A few detailed comments

Response:

 In the revised conclusion section, we have incorporated detailed comments in red font.

 Line 486 Conclusion in revision document:

   By using gas chromatography coupled with ion mobility spectrometry, we established fingerprint profiles of koumiss from four different regions. Combination of the application of PLS-DA analysis, we identified dimethyl sulfide, ethyl butyrate, 2-methylbutanal, ethyl propionate, 2-methylpropyl propionate, isobutyl acetate, n-propanol, and methyl ethyl ketone as indicative markers for the rapid detection of traditionally fermented koumiss from these regions. In conclusion, it is worth emphasizing that the use of GC-IMS makes it possible to identify regions of origin, and thus control the authenticity of food origin. This is important for different food groups, but for regional and traditional food in particular. By utilizing GC-IMS and GC-MS to analyze the volatile components of the koumiss from these regions, we found that the volatile components were similar in terms of variety. However, due to differences in the fermentation process between regions, the volatile components exhibited significant differences in concentration and displayed clear regional characteristics. A total of 87 volatile components were detected, with esters, acids, and alcohols being the main aromatic substances in the koumiss. Furthermore, a comparison of these two detection methods regarding volatile compound analysis revealed that GC-IMS exhibited a more sensitive response towards aldehydes and ketones.Subsequently, an analysis of the OAV for koumiss from different regions identified ethyl caprylate, ethyl caprate, octanoic acid, 9-decenoic acid ethyl ester, n-decanoic acid, isopentanol, phenethyl alcohol, and ethyl hexanoate as key aroma components in traditional fermented koumiss of the Kazakh ethnic group. Interestingly, the aroma profile of the ALTw region was primarily dominated by phenethyl alcohol, while the YL and TC regions showcased a prominent presence of ethyl caprylate and ethyl caprate. Additionally, a qualitative descriptive analysis (QDA) was conducted to evaluate and categorize the aroma profiles of koumiss from the four regions, resulting in three distinct aroma types: fruit-cream type, milk type, and floral type.

3.Comment:

  Figure 1 a and b - improve quality

Response:

The overall quality of Image 1 has been enhanced from 403k to 14.5m.Line 111 in the revision document.

4.Comment:

Line 339 Cut the text and insert the caption of table No. 3

Response:

We sincerely apologize for the oversight in applying the formatting template, resulting in the omission of the title for Table 3.The necessary modifications have now been completed. Line 412 in revision document .Table 3: OVA content of volatile substances in koumiss from different regions

5.Comment:

Suggestions for the future:It seems that it would be very interesting to analyze the composition of koumiss produced in individual regions and coming from several producers. It would then be possible to answer the question of whether it is possible to determine the diversity of kumiss within one region?

Response:

  Thank you for your valuable suggestions regarding our research. Analyzing the diversity of koumiss in the same region is indeed a worthwhile experiment for future studies. We are now prepared to commence the analysis of koumiss diversity in the same region.

Thank you and best regards.

Yours sincerely,

Gou Yongzhen

Xu Qian

E-mail: xuqiantaru@126.com

Reviewer 2 Report

Please find some comments and suggestions inserted in the manuscript 

Author Response

Dear Dr

Thank you for the reviewers' comments concerning our manuscript entitled "Exploring the Characteristic Aroma Components of Traditional Fermented koumiss of Kazakh Ethnicity in Different Regions of Xinjiang by Combining Modern Instrumental Detection Technology with Multivariate Statistical Analysis Methods for Odor Activity Value and Sensory Analysis" (ID: 2403182). Those comments are all valuable and very helpful for revising and improving our paper, as well as the important guiding significance to our researches. We have studied comments carefully and have made changes which we hope meet with approval. Revised portion are marked in red in the attachment.The revised document is saved in the attachment.

Here is a point-by-point response to the comments and concerns of the reviewers

Comment: “please start the sentence use the capita;please mention the city and the country of the reagent's producers;please re-check all the unit, not all the units were correctly written;supposed to be "analysis";the significance level should be written in italic form;the scientific name should be written in a italic form”

Response:We deeply apologize for any inconvenience caused by the writing errors that hindered your reading experience. We have made the necessary corrections to the writing errors in the article and highlighted them in red in the attached document.

Comment:Line 43 in the modification document“what koumiss is has been explained at line 32, so this sentence is not necessary and can be removed”

Response:We have rewritten this section based on your suggestion:“In recent years, with the burgeoning development of the equine industry in Xinjiang, koumiss, as a high-value-added food, has gradually transformed from a mere pastoral self-sustenance beverage to a constantly selling commodity”.Duplicate parts were removed,in the 42 lines in the revision document。

Comment:Line 86 in the modified opinion document:“according to the previous paragraph, GC MS has been used to detect the volatile compound. please give a justification why in this research was utilized the GC-IMS also MS?”

Response:We have rewritten this section based on your suggestion:“This study focuses on the traditional fermented koumiss of the Kazakh ethnic group in four different regions of Xinjiang. By utilizing the convenient and rapid GC-IMS technique and employing multivariate statistical analysis, we have established fingerprint profiles to differentiate the volatile compounds of koumiss from different regions. In tandem with GC-MS analysis, utilizing volatile compounds as the foundation, we have integrated odor activity value (OAV) analysis and quantitative descriptive analysis (QDA) based on sensory evaluation to characterize the sensory aromatic attributes of koumiss from various regions.”The 85 lines in the revision document. Considering the convenience and rapid generation of fingerprint profiles offered by GC-IMS, we initially planned to utilize this method in our experimental design for distinguishing koumiss varieties from different regions. However, after conducting thorough research and preliminary experiments, we found that GC-IMS is not optimal for the comprehensive analysis of volatile compounds. Consequently, we have made the final decision to combine GC-MS and GC-IMS in our experiments, which will also allow us to examine and compare the differences between GC-IMS and GC-MS analyses.

Comment:Line 97 in the modified opinion document:“please give a brief explanation about regarding the sampling consideration carried out”

Response:We have rewritten this section based on your suggestion:“To ensure the representative significance of samples from different regions, the linear distance between sampling locations is guaranteed to exceed 350 kilometers.” The 100 lines in the revision document.

Comment:Line 237 in the modified opinion document:“what is the meaning of green, yellow, and red box on the list of the compounds?”

Response:We have rewritten this section based on your suggestion:“The fingerprint spectra of the volatile constituents in koumiss from distinct regions.The delineated color zones represent the variances in volatile substances among different regions.” The 246 lines in the revision document. Specific analysis is performed in lines 227 to 239.

Comment:Line 281 in the modified opinion document:“why this happened?”

Response:We have rewritten this section based on your suggestion:“Although IMS has a strong sensitivity to the ion drift of volatile compounds without fragmentation, and can identify 76 peaks, 26 volatile compounds cannot be confirmed due to incomplete databases.” The 293 lines in the revision document.

Comment:Line 399 in the modified opinion document“please change the title related to the presented data”

Response:We sincerely apologize for the oversight in applying the formatting template, which resulted in the omission of the title for Table 3. The necessary modifications have now been completed:“Table 3 : OVA content of volatile substances in koumiss from different regions.” In the 412 lines in the modified document.

Comment:Line 414 in the modified opinion document“what OAV stands for?”

Response:We will add a part prior to this sentence according to your suggestion:“Odor Activity Value (OAV) refers to the ratio between the mass concentration of an aroma compound and its sensory threshold in an odor system.” In the 425 lines in the modified document.

Comment:Line 420 in the modified opinion document:“please mention which compound indicated as a key marker for each types of kumiss”

Response:In the revised document, lines 430 to 450 explain that ethyl caprate and ethyl caprylate are the primary aroma compounds in the YL region, while phenethyl alcohol is the main aroma compound in the ALTe region.

Comment:Line 461 in the modified opinion document:“please remove this sentence”

Response:We sincerely apologize for the oversight in applying the text formatting template, which resulted in the omission of deleting that sentence. It has been deleted.

Comment:Line 488 in the modified opinion document:“were those detected compound as the key aromatic component of Kazakh koumiss in comparison with other types of koumiss or as a key marker for the four types of the sample used?”

Response:The aforementioned compounds are key aroma compounds in Kazakh traditional koumiss and greatly impact the sensory experience. However, the emphasis on these compounds varies in different regions. For instance, the aroma profile of the ALTw region is primarily dominated by phenethyl alcohol, with other compounds playing a secondary role. On the other hand, the aroma profiles of the YL and TC regions are primarily characterized by ethyl caprate and ethyl decanoate, with other compounds playing a secondary role.We have rewritten this section based on your suggestion:“Subsequently, an analysis of the OAV for koumiss from different regions identified ethyl caprylate, ethyl caprate, octanoic acid, 9-decenoic acid ethyl ester, n-decanoic acid, isopentanol, phenethyl alcohol, and ethyl hexanoate as key aroma components in traditional fermented koumiss of the Kazakh ethnic group. Interestingly, the aroma profile of the ALTw region was primarily dominated by phenethyl alcohol, while the YL and TC regions showcased a prominent presence of ethyl caprylate and ethyl caprate. ”In the 503 lines in the modified document.

Comment:Line 500 in the modified opinion document:“human was equipped for sensory analysis. the author should provide the information of the ethical clearance”

Response:We have rewritten this section by following your suggestion:“This article encompasses human olfactory sensory experiments, conducted with the necessary consent obtained from all participants, along with appropriate training provided to the participants. The article does not involve experimental research on animal subjects.”In the 522 lines in the modified document.

Thank you and best regards.

Yours sincerely,

Gou Yongzhen

Xu Qian

E-mail: xuqiantaru@126.com

Round 2

Reviewer 2 Report

Most of the reviewer’s comments have been accommodated by the author. However, minor points need to be re-check before considered for publication. Those are listed below: 

- line 61: the “subsp.” for the scientific name should not be in the italic form

- line 40: please start the sentence using capital